# Species Distribution, Antimicrobial Resistance, and Enterotoxigenicity of Non-*aureus* Staphylococci in Retail Chicken Meat

**DOI:** 10.3390/antibiotics9110809

**Published:** 2020-11-13

**Authors:** Soo In Lee, Sun Do Kim, Ji Heon Park, Soo-Jin Yang

**Affiliations:** Department of Animal Science and Technology, School of Bioresources and Bioscience, Chung-Ang University, Anseong 17546, Korea; duddj1101@cau.ac.kr (S.I.L.); seindo@cau.ac.kr (S.D.K.); pjheon26@cau.ac.kr (J.H.P.)

**Keywords:** non-*aureus* staphylococci, chicken meat, antimicrobial resistance, fluoroquinolone resistance, staphylococcal enterotoxin

## Abstract

Non-*aureus* staphylococci (NAS), including coagulase-negative staphylococci, have emerged as important causes of opportunistic infections in humans and animals and a potential cause of staphylococcal food poisoning. In this study, we investigated (i) the staphylococcal species profiles of NAS in in retail chicken meat, (ii) the phenotypic and genotypic factors associated with antimicrobial resistance in the NAS isolates, and (iii) the prevalence of classical and newer staphylococcal enterotoxin (SE) genes. A total of 58 NAS of nine different species were isolated from retail raw chicken meat samples. The occurrence of multidrug resistance in the NAS, particularly *S. agnetis* and *S. chromogenes*, with high resistance rates against tetracycline or fluoroquinolones were confirmed. The tetracycline resistance was associated with the presence of *tet*(L) in *S. chromogenes* and *S. hyicus* or *tet*(K) in *S. saprophyticus*. The occurrence of fluoroquinolone resistance in *S. agnetis* and *S. chromogenes* was usually associated with mutations in the quinolone resistance determining regions (QRDR) of *gyrA* and *parC*. In addition, the frequent presence of SE genes, especially *seh*, *sej*, and *sep*, was detected in *S. agnetis* and *S. chromogenes*. Our findings suggest that NAS in raw chicken meat can have potential roles as reservoirs for antimicrobial resistance and enterotoxin genes.

## 1. Introduction

Staphylococci are commensal colonizers of the skin and mucous membranes of humans and various animals [1,2,3]. Staphylococci are typically divided into coagulase-positive (CoPS) or coagulase-negative staphylococci (CoNS) according to their ability to produce coagulase [4,5]. *Staphylococcus aureus* has been the only well-known CoPS that is recognized as a major human and animal pathogen with many virulence factors [6,7,8]. *S. aureus* can cause an array of local and systemic infections in human and animal hosts, and is among the most prevalent community- and health care-associated human pathogens [8]. Although usually self-limiting, staphylococcal food poisoning (SFP) has been reported to be one of the most common food-borne diseases that results from the ingestion of food contaminated with staphylococcal enterotoxins (SEs), which are commonly secreted by *S. aureus* [3,9]. In addition to the various virulence factors and enterotoxigenicity in staphylococci, antimicrobial-resistant staphylococcal isolates, especially methicillin-resistant staphylococci (MRS), have been increasingly reported in recent years [10,11]. The methicillin resistance phenotype in staphylococci is mostly caused by the *mecA* gene that is located within a staphylococcal cassette chromosome *mec* (SCC*mec*) [12]. In addition to *mecA*, other *mec* genes, such as *mecB* and *mecC*, have also been recognized in association with β-lactam resistance in staphylococci [13,14]. The *mecB* and *mecC* genes have usually been identified within mobile genetic elements (MGEs) that were similar to SCC*mec* [15]. Several studies have reported that these SCC*mec* elements can be transferred between CoPS and CoNS isolates [16,17]. Moreover, the frequent occurrence of fluoroquinolone resistance, as a result of mutations in quinolone resistance determining regions (QRDR), has been observed in the clinical isolates of multidrug-resistant methicillin-resistant *S. aureus* (MDR-MRSA) and CoNS [18,19,20].

Recently, non-*aureus* staphylococci (NAS), including CoNS, have emerged as important causes of opportunistic infections in humans and animals and a potential cause of food poisoning [1,2]. The presence of genes encoding various SEs and toxic shock syndrome toxin-1 (TSST-1) has also been reported in CoNS and NAS isolates from human [21,22], animal [23,24], and retail meat samples [6,25] despite of their controversial pathophysiological role in SFP. The importance of foods of animal origin as carriers of foodborne pathogens has been demonstrated and retail chicken meat has been recognized as one of the significant food vehicles of zoonotic pathogens and antimicrobial resistance, as well as staphylococcal enterotoxin genes [6,10,26]. Many countries, including European and North American countries, routinely monitor and publish surveillance reports on antimicrobial-resistant pathogens in chicken and retail chicken meat that both usually include *S. aureus* (MRSA and MSSA) [27,28].

While CoPS, especially *S. aureus* isolates, have been extensively investigated as a major causative agent of SFP, detailed data on the prevalence of CoNS or NAS in foods of animal origin, their antimicrobial resistance profiles, and enterotoxigenic profiles have been limited. Although a detailed pathogenic role of SE-producing CoNS or NAS in SFP has not yet been elucidated, recent studies have reported that some CoNS can produce enterotoxins in a range of ng/mL [29]. In addition, many recent publications highlighted the potential involvement of CoNS or NAS in human and animal diseases [1,2,30]. Thus, we aimed to investigate (1) the species profiles of NAS in retail chicken meat samples that were collected in Korea, (2) the antimicrobial resistance profiles of the NAS isolates and genetic factors associated with the antimicrobial-resistant phenotype, and (3) the prevalence and distribution of five classical SE genes (*sea, seb, sec, sed, see*), 13 newer SE genes (*seg, seh, sei, selj, sek, sell, sem, sen, seo, sep, seq, ser, selu*), and toxic shock syndrome toxin-1 gene (*tst1*) in NAS isolates.

## 2. Results

### 2.1. Profiles of NAS Isolated from Retail Chicken Meat

A total of 58 NAS isolates of nine different species were isolated from 128 retail chicken meat samples (58/128, 45.3%) that were collected in 2019. All the 58 NAS used in this study were isolated from different samples. Based on previous publications that reported the coagulase-variable phenotype in *S. hyicus*, *S. chromogenes*, and *S. agnetis* isolates, these species were grouped as coagulase-variable staphylococci (CoVS) in this study [31,32,33]. As shown in Table 1, 29 of the 58 NAS were coagulase-variable staphylococci (CoVS), and the other 29 NAS were CoNS. The CoVS and CoNS strain groups had three and six different species of staphylococci, respectively. The most common NAS species that were isolated from retail chicken meat were *S. agnetis* (*n* = 12, 19.4%), *S. saprophyticus* (*n* = 11, 19%), *S. chromogenes* (*n* = 9, 14.5%), *S. hyicus* (*n* = 8, 12.9%), and *S. sciuri* (*n* = 8, 13.8%).

### 2.2. Occurrence of mecA in NAS Isolated from Retail Chicken Meat

Three *mecA*-positive strains were identified in the 58 NAS (5.2%) isolated from the retail chicken meat samples (Table 1). These three isolates (two *S. sciuri* and one *S. lentus*) were all CoNS and exhibited an OXA-resistant phenotype (OXA MICs ≥ 0.5 µg/mL). The three *mecA*-positive strains were subjected to SCC*mec* typing analysis; no SCC*mec* types were determined, as *ccr* genes were not detected.

### 2.3. Antimicrobial Resistance Profiles of NAS Isolated from Retail Chicken Meat

All the 58 NAS isolates exhibited susceptibilities to cefoxitin (FOX), quinupristin-dalfopristin (SYN), vancomycin (VAN), and linezolid (LZD) (Table 2 and Table 3). The CoVS strains tended to display higher resistance than the CoNS strains to the antimicrobial agents tested, with exception to mupirocin (MUP) and rifampin (RIF). A higher level of multidrug resistance (MDR) phenotype (resistant to ≥3 subclasses of antimicrobial drugs) was also observed in the CoVS strains (48.3%) than in the CoNS (10.3%) strains (Table 2 and Table 3). The CoVS strains displayed rather high levels of resistance to tetracycline (TET, 48.3%) and fluoroquinolones (55.2%). The CoNS strains also showed 34.5% of TET resistance. Interestingly, all the 14 TET-resistant CoVS possessed *tet*(L) (Table 2 and Table 3). On the other hand, all the 10 TET-resistant CoNS harbored *tet*(K), and one SM1 strain also harbored *tet*(M) (Table 2 and Table 3). 

### 2.4. Fluoroquinolone Resistance and Mutations in the QRDR

Among the CoVS strains, seven of 12 *S. agnetis* (58.3%) and all nine *S. chromogenes* (100%) exhibited resistance to three fluoroquinolones (CIP, ENR, and LEV) (Table 2). On the contrary, only two strains of CoNS (one *S. simulans* and one *S. lentus*) were resistant to the fluoroquinolones. Since fluoroquinolone resistance in staphylococci has been known to be associated with mutations in the QRDRs encoding DNA gyrase and topoisomerase IV [19,20], we examined whether mutations in the QRDR of *gyrA*, *gyrB*, *parC*, and *parE* of the 18 fluoroquinolone-resistant NAS strains were present.

As shown in Table 4, S84L mutation was identified in the *gyrA* of all 18 fluoroquinolone-resistant strains. S80L mutation within *parC* was observed in 17 strains; S80L mutation was not observed in *S. simulans* strain (SM4). Moreover, mutations at codon 84 in *parC* (D84E, D84H, D84G, D84Y, or D84N) were associated with CIP or ENR resistance at higher concentrations (CIP MICs ≥ 64 or ENR MICs ≥ 128 µg/mL). Although four strains of *S. agnetis* (SA4, SA5, SA11, and SA12) had two or more additional point mutations at codons 424, 426, 465, 488, 489, or 491 in *parE* genes, the four *S. agnetis* strains did not show markedly higher fluoroquinolone resistance than the other fluoroquinolone-resistant strains without the mutations (Table 4). Among the 18 fluoroquinolone-resistant strains, only one *S. agnetis* strain (SA6) had a point mutation in *gyrB*. 

### 2.5. Occurrence and Distribution of SE Genes in NAS

While only two of the 29 CoVS strains (6.9%) were negative for all the SE genes detected, 15 of the 29 CoNS strains (51.7%) did not possess any SE gene (Table 3 and Table 5). Overall, CoVS strains (*S. agnetis*, *S. chromogenes*, and *S. hyicus*) harbored a great number of SE genes than the six species of CoNS strains. Of the 29 CoVS strains, 16 (55.2%) possessed three or more SE genes, and 10 CoVS (34.5%) possessed the *tst1* gene (Table 5). Unlike the CoVS, none among the 29 CoNS strains possessed more than two SE genes, and only two CoNS strains (6.9%) had the *tst1* gene.

Among the five classical and 13 newer SE genes, *seh* (20/29, 70%), *sep* (13/29, 44.8%), *selj* (13/29, 44.8%), and *seo* (8/29, 27.6%) genes were the most frequently detected in CoVS strains. Furthermore, all *S. agnetis* strains were double positive for *sep* and *seh*, except the SA4 strain (Table 3 and Table 5). However, one of the five classical SE genes (*see*) and two of the 13 newer SE genes (*sell* and *seq*) were found in only CoNS species such as *S. saprophyticus*, *S. sciuri*, *S. lentus*, or *S. warneri* (Table 5). Likewise, *sea* and *sem* genes were identified in only three species of CoVS, *S. agnetis*, *S. chromogenes*, or *S. hyicus*.

## 3. Discussion

The contamination of raw food products with pathogens poses a significant threat to public health and has caused a great burden in health care. SFP is one of the most common foodborne illnesses in many countries and is caused by the ingestion of foods contaminated with SEs [9]. Since staphylococci are usually found on the skin and the mucous surface of the respiratory tract of human and animal hosts, foods of animal origin such as milk, cheese, pork, beef, and poultry meat have been recognized as a major source of SFP as a result of contamination with staphylococci, especially coagulase-positive *S. aureus* [4]. Although *S. aureus* has been well recognized for its ability to evoke SFP via the production of enterotoxins that may have emetic and superantigenic abilities [9], only a limited number of studies have focused on the significance of non-*aureus* CoPS and CoNS in food safety and public health.

In addition to their enterotoxigenic ability, staphylococci are able to develop resistance to various antimicrobials through different genetic mechanisms [20,34]. In particular, livestock-associated methicillin-resistant *S. aureus* (LA-MRSA) has been detected in many raw meat products [6] including pork, beef, lamb, and poultry [27,35,36]. Recent reports also highlighted the importance of CoNS and NAS as potential reservoirs of antimicrobial resistance genes [16,26]. Of note, it has been reported that food-originated staphylococci carrying SCC*mec* and multiple antimicrobial resistance genes in the food production chain could be a substantial reservoir for transmission of the resistance genes [6,10,37,38]. Previous studies also demonstrated that plasmids carrying various antimicrobial resistance genes such as *cfr*, *erm*(C), *erm*(T), *lnu*(A), or *dfrK* were identified in both MRSA and CoNS, indicating horizontal transmission of the plasmids across bacterial species in various environment [39,40,41]. Furthermore, Cuny et al. reported occurrence of *cfr*-carrying plasmids in CoNS isolates from calves and veterinarians along with the transferability of the plasmids among different staphylococcal species [42].

In this study, we assessed the distribution and species profiles of NAS in retail chicken meat samples that were purchased from eight different provinces of Korea. The overall prevalence rate of NAS in retail chicken meat samples was 45.3% and 11 different species of NAS were identified from the retail chicken meat samples (Table 1). Previous investigations that were conducted in Turkey, Egypt, and Brazil also revealed the presence of various coagulase-positive or coagulase-negative NAS in retail chicken meat [6,37,38]. In particular, Osman et al. [38] reported nine different species of NAS, e.g., *S. lugdunensis* (30%), *S. epidermidis* (26%), and *S. hyicus* (20%) were the three most frequently isolated NAS from retail chicken meat in Egypt [38]. The most frequently detected species of NAS in our investigation were *S. agnetis* (19.4%) and *S. saprophyticus* (19%) and these differences might have been caused by various factors, including isolation and enrichment methods, geographic region, and differences in chicken meat production. Recently, the involvement of NAS, such as *S. epidermidis*, *S. saprophyticus*, and *S. haemolyticus* in human infections, have been highlighted in several publications [1,2]. Moreover, *S. chromogenes* and *S. simulans* have been found in a number of animal infections, especially bovine mastitis [30]. These findings suggest that the frequent occurrence of NAS in raw chicken meat may be a significant hazard associated with food and public health safety.

The prevalence of antimicrobial-resistant NAS in retail chicken meat has not yet been well investigated. Recent reports on methicillin-resistant NAS, especially those of CoNS species, in many livestock and companion animals have raised concern regarding the unattended transmission of NAS to human through the food production chain [7,10]. In the present study, only three (SI1, SI2, and SL2) among the 58 NAS isolates were positive for *mecA* gene and exhibited OXA resistance phenotype (OXA MIC ≥ 0.5 µg/mL) (Table 1). Moreover, a strain of *mecA*-positive *S. lentus* (SL2) exhibited the highest resistance to OXA (OXA MIC of 16 µg/mL) (Table 3). Furthermore, the SCC*mec* type of the three *mecA*-positive NAS (SI1, SI2, and SL2) were not identified. In line with our results, previous studies also reported non-typeable *ccr* genes associated with heterogeneity of SCC*mec* elements in methicillin-resistant CoNS strains [12,43]. Although 22 NAS strains (four *S. agnetis*, two *S. chromogenes*, 10 *S. saprophyticus*, four *S. sciuri*, one *S. lentus*, and one *S. warneri*) displayed OXA resistance phenotype, *mecA* was not detected in these strains. Thus, based on the CLSI guidelines [44,45], the 22 strains were determined as methicillin-susceptible staphylococci. As presented in Table 2, MDR phenotype was widespread in *S. agnetis* and *S. chromogenes* strains. These results indicate that non-*aureus* CoVS and CoNS can serve as potential reservoirs for antimicrobial resistance genes and necessitate further attention.

Staphylococcal strains isolated from raw meat samples have been reported to have a high level of TET resistance rates (>62%) [28,37,38]. In line with these studies, we observed a relatively high level of TET resistance rate (41.1%) in the NAS isolates (Table 2). This was mainly attributed to *S. chromogenes*, *S. hyicus*, and *S. saprophyticus*, which showed TET resistance rates of 66.7%, 100%, and 63.6%, respectively (Table 2). TET resistance is usually conferred by the acquisition of *tet*(M), *tet*(O), and *tet*(S), which encode ribosomal protection proteins, or *tet*(K) and *tet*(L), which encode efflux pumps [34]. Similar to previous publications [26,46], all of the TET-resistant CoNS including *S. saprophyticus*, *S. sciuri*, *S. simulans*, and *S. lentus* possessed *tet*(K). In contrast to the TET-resistant CoNS, all the *S. chromogenes* and S*. hyicus* strains with the TET-resistant phenotype possessed *tet*(L). Unlike the chromosomal or transposonal location of *tet*(M) or *tet*(O), plasmid-borne *tet*(K), and *tet*(L) genes [34] might have contributed to high incidence of TET-resistant NAS strains in this study.

Quinolones exert strong antibacterial activity through its action against DNA gyrase and topoisomerase IV, which are essential enzymes in bacteria. However, the frequent occurrence of quinolone resistance has been reported in clinical isolates of MRSA and CoNS through point mutations in the genes encoding the two essential enzymes [19,20]. In this study, sequencing analyses of the QRDR region of *gyrA*, *gyrB*, *parC*, and *parE* genes revealed that the fluoroquinolone resistance observed in seven of 12 *S. agnetis* and all nine *S. chromogenes* isolates were associated with mutations in *gyrA* at codon 84 and *parC* at codon 80 or 84 (Table 4), which is similar to what was observed in fluoroquinolone-resistant *S. aureus* [20]. The two CoNS (SM4 and SL2) strains also had the same mutations in the QRDR of *gyrA* and *parC*. The widespread usage of fluoroquinolones in the poultry industry has been associated with the emergence and prevalence of the fluoroquinolone-resistant bacteria. In 2018, Korea Animal Health Products Association (KAHPA) reported that quinolones and tetracyclines were the two antimicrobials that were sold in the largest amount for use in the poultry industry [47]; in line with this report the number of NAS strains with resistance to tetracycline and fluoroquinolones was observed the highest in the present study (Table 2).

Although not approved for use in veterinary medicine, the occurrence of linezolid- or vancomycin-resistant staphylococci in food-producing animals has caused serious public health problems [6,48]. The presence of *cfr*, *optrA*, and *vanA* genes in staphylococci has frequently been associated with high level resistance to linezolid and vancomycin, respectively [49,50]. Fortunately, none of the 58 NAS strains showed resistant phenotype to the two critically important antimicrobial agents (Table 3). However, nationwide surveillance of the linezolid- and vancomycin-resistant staphylococci in livestock is necessary for a future investigation.

Classical and newer SEs have been associated mainly with SFP cases involving coagulase-positive *S. aureus*. However, the presence of SE genes in NAS such as *S. epidermidis*, *S. hyicus*, *S. haemolyticus*, and *S. chromogenes* has recently been reported [6,51]. In the present study, we found that *S. agnetis* and *S. chromogenes* species from retail chicken meat samples often carried multiple newer SE genes (Table 2 and Table 5). In contrast to the previous publications that reported the highest prevalence of *sec* [51,52] in CoNS, *seh*, *selj*, and *sep* were most frequently detected in the *S. agnetis* and *S. chromogenes* isolates in the present study. Although the role of newer SEs in outbreak of SFP is still controversial, an outbreak case of SFP caused by SEH produced by *S. aureus* has been reported in Norway [53]. Johler et al. (2015) also suggested that newer SE genes, such as *seg*, *sei*, *sem*, *sen*, and *seo*, may have caused an SFP outbreak in Switzerland [54]. Despite the pathophysiological role of SEs in NAS remains controversial and the profiles of SE genes in different NAS species are variable, the substantial presence of SE genes and TSST-1 gene in NAS isolated from retail chicken meat may pose a significant hazard to food safety. The production of classical SEs is affected by many conditions such as genetic factors of the strains, culture environment, cell density, and changes in bacterial cell membrane physiology [9]. Since data on the regulation of newer enterotoxins are limited even in the well-recognized *S. aureus*, further research is warranted to identify the role and expression profiles of SE genes that are detectable in NAS, i.e., *seh, sei,* and *sep* genes in *S. agnetis* and *S. chromogenes*.

It should be recognized that our results in the current study were generated from a rather limited number of retail chicken meat samples and NAS isolates. In addition, prevalence of *mecB* and *mecC* in the NAS was not included in the current study. Furthermore, detailed molecular mechanisms involved in resistance to chloramphenicol, erythromycin, and clindamycin were not characterized. Nonetheless, future studies should be focused on complete genotypic and phenotypic characterizations of NAS for virulence, production of SEs, and antimicrobial resistance mechanisms. Moreover, the present study is the first to report profiles of antimicrobial resistance and enterotoxigenicity in NAS collected from retail chicken meat samples in Korea.

## 4. Materials and Methods

### 4.1. Sample Collection

Retail chicken meat samples (breasts and thighs, *n* = 128) were collected from 20 retail markets and groceries in eight provinces of Korea during 2019. The prepackaged chicken meat samples were placed in a container with ice packs to keep the samples below 4 °C and sent to our laboratory for the isolation of staphylococci within 24 h of sampling.

### 4.2. Isolation and Identification of Staphylococci

Chicken meat samples weighing 25 g were homogenized for 2 min in 225 mL buffered peptone water in a sterile stomacher bag (3M, St. Paul, MN, USA) using a HAPS^®^ homogenizer H3-1 (HUKO, Seoul, Korea). Then, 1 mL aliquot of the homogenized solutions were inoculated into 9 mL of fresh Tryptic Soy Broth (TSB, Difco Laboratories, Detroit, MI, USA) containing 10% NaCl and enriched for 24 h at 37 °C. Next, 20 μL aliquots of the pre-enriched cultures were streaked onto Baired–Parker Agar (BPA; Difco Laboratories) supplemented with egg yolk and potassium tellurite, and then incubated at 37 °C for 48 h. Presumptive staphylococcal colonies were selected, and 2–3 colonies of the most dominant colony type from each sample were streaked on BPA for subsequent identification. Individual colonies were inoculated into fresh TBS, and incubated for 18–24 h, and genomic DNA from bacterial cell pellets were extracted using a Genmed DNA kit (Seoul, Korea) according to the manufacturer’s recommendations. The identification of staphylococcal species was performed by using a 16s rRNA sequencing method as previously reported [55].

### 4.3. Antimicrobial Susceptibility Tests

Antimicrobial susceptibility assay was performed on all staphylococci according to the Clinical and Laboratory Standards Institute guidelines [44]. Sixteen antimicrobial agents were utilized for disc diffusion assays on Mueller–Hinton agar (MHA, Difco Laboratories): ampicillin (AMP, 10 μg), cefoxitin (FOX, 30 μg), penicillin (PEN, 10 μg), chloramphenicol (CHL, 30 μg), ciprofloxacin (CIP, 5 μg), enrofloxacin (ENR, 5 μg), levofloxacin (LEV, 5 μg), clindamycin (CLI, 2 μg), erythromycin (ERY, 15 μg), gentamicin (GEN, 10 μg), mupirocin (MUP, 200 μg), rifampicin (RIF, 5 μg), sulfamethoxazole-trimethoprim (SXT, 23.75/1.25 μg), quinupristin-dalfopristin (SYN, 15 μg), and tetracycline (TET, 30 μg). Mupirocin was purchased from Oxoid (Hampshire, UK), and the rest of the antimicrobial agents were purchased from BD BBL^TM^ (Becton Dickinson, Franklin Lakes, NJ). The minimum inhibitory concentrations (MICs) of oxacillin (OXA), TET, CIP, and ENR were determined for all the study strains by using the standard two-fold broth microdilution [44]. The MICs of the study strains to vancomycin (VAN) and linezolid (LZD) were determined by standard Etest (AB Biodisk, Dalvagen, Sweden). *S. aureus* MW2 and *S. aureus* ATCC 29213 strains were used as reference strains for the antimicrobial susceptibility tests. All antimicrobial susceptibility tests were repeated three times.

### 4.4. Detection of Antimicrobial Resistance Genes and SCCmec Typing

The staphylococcal strains that exhibited OXA, FOX, or TET resistance phenotypes were examined for the presence of resistance genes. MRS strains were examined for the presence of *mecA*, and SCC*mec* types were determined as previously described [56,57]. For SCC*mec* typing, multiplex PCR methods were used to amplify chromosomal cassette recombinase (*ccr*) genes and *mec* regulatory elements (*mec*). The combinations of *ccr* types and *mec* complexes were used to define the SCC*mec* element types of staphylococcal strains. TET-resistant staphylococcal strains were examined for the carriage of *tet*(K), *tet*(L), *tet*(M), *tet*(O), and *tet*(S) using specific primer sets as previously described [58].

### 4.5. Detection of Mutations in QRDRs

The two primary targets of fluoroquinolones are bacterial DNA gyrase and topoisomerase IV. In most fluoroquinolone-resistant staphylococcal species, point mutations occur in highly-conserved QRDRs encoding the DNA gyrase (*gyrA* and *gyrB*) and topoisomerase IV (*parC* and *parE*) [59]. The genomic DNA samples of NAS strains were subjected to PCR amplifications using the specific primer sets as shown in Appendix A as previously described [18,19,20]. The published sequences of *S. agnetis* (908, NCBI GenBank accession number CP009623), *S. chromogenes* (17A, NCBI GenBank accession number CP031274), *S. simulans* (FDAARGOS_124, NCBI GenBank accession number CP14016), and *S. lentus* (NCTC12102, NCBI GenBank accession number UHDR01000002) were used as reference for the design of the gene-specific primer sets. The resulting amplicons were sequenced at Cosmo Genetech, Seoul, Korea. Multiple *gyrA*, *gyrB*, *parC*, and *parD* sequence alignments were performed using the Box-Shade server (embnet.vital-it.ch/software/BOX_form.html).

### 4.6. Detection of SE Genes

The detection of five classical SE genes (*sea*, *seb*, *sec*, *sed*, *see*), 13 newer SE genes (*seg*, *seh*, *sei*, *selj*, *sek*, *sell*, *sem*, *sen*, *seo*, *sep*, *seq*, *ser*, *selu*), and the TSST-1 gene (*tst1*) in the staphylococcal strains was performed using a series of multiplex PCR assays as previously described [60,61]. The genomic DNA samples from reference *S. aureus* strains (N315, FRI472, MW2, FRI913, and COL) were used as positive control samples for each PCR assay.

## 5. Conclusions

In conclusion, our results demonstrate that (1) a relatively high level of diverse species of NAS are present in retail raw chicken meat; (2) the occurrence of MDR in the NAS isolates, particularly *S. agnetis* and *S. chromogenes*, with high resistance rates against TET and/or fluoroquinolones were observed; (3) the TET resistance phenotype was associated with the presence of *tet(L)* or *tet(K)*; (4) the prevalent occurrence of fluoroquinolone-resistant *S. agnetis* and *S. chromogenes* was caused by mutations in the QRDR of *gyrA* and *parC*; and (5) the presence of newer SE genes such as *seh*, *selj*, and *sep*, in addition to antimicrobial resistance, were detected in *S. agnetis* and *S. chromogenes*. Our results suggest that the presence of NAS, which have pathogenic potential and are reservoirs of antimicrobial resistance and enterotoxin genes, in raw chicken meat should not be overlooked.

## Figures and Tables

**Table 1 antibiotics-09-00809-t001:** Profiles of non-*aureus* staphylococci (NAS) isolated from retail chicken meat in Korea.

NAS	No. of *mecA*- Positive Strains (%, Type of SCC*mec*)
CoVS (*n* = 29, 50%)	
*S. agnetis* (*n* = 12, 20.7%)	-
*S. chromogenes* (*n* = 9, 15.5%)	-
*S. hyicus* (*n* = 8, 13.8%)	-
CoVS Total	-
CoNS (*n* = 29, 50%)	
*S. saprophyticus* (*n* = 11, 19%)	-
*S. sciuri* (*n* = 8, 13.8%)	2 (25, NT, NT)
*S. simulans* (*n* = 5, 8.6%)	-
*S. lentus* (*n* = 2, 3.4%)	1 (50, NT)
*S. warneri* (*n* = 2, 3.4%)	-
*S. epidermidis* (*n* = 1, 1.7%)	-
CoNS Total	3 (10.3)
TOTAL	3 (5.2)

NAS, non-*aureus* staphylococci; CoVS, coagulase-variable staphylococci; CoNS, coagulase-negative staphylococci; NT, non-typable.

**Table 2 antibiotics-09-00809-t002:** Antimicrobial resistance profiles of 58 non-*aureus* staphylococci (NAS) strains isolated from retail chicken meat.

NAS (*n* = Isolates)		No. of Antimicrobial Resistance (%)													
		AMP	FOX	PEN	CHL	FQN	CLI	ERY	GEN	MUP	RIF	SXT	SYN	TET	MDR ^1^
CoVS
	*S. agnetis* (12)	6 (50)	-	6 (50)	3 (25)	7 (58.3)	4 (33.3)	4 (33.3)	3 (25)	-	-	-	-	-	5 (41.7)
	*S. chromogenes* (9)	4 (44.4)	-	4 (44.4)	3 (33.3)	9 (100)	8 (88.9)	8 (88.9)	-	-	-	2 (22.2)	-	6 (66.7)	9 (100)
	*S. hyicus* (8)	-	-	-	-	-	-	-	-	-	-	-	-	8 (100)	0
	CoVS Total (29)	10 (34.5)	-	10 (34.5)	6 (20.7)	16 (55.2)	12 (41.4)	12 (41.4)	3 (10.3)	-	-	2 (6.9)	-	14 (48.3)	14 (48.3)
CoNS
	*S. saprophyticus* (11)	-	-	-	-	-	-	-	-	-	-	-	-	7 (63.6)	-
	*S. sciuri* (8)	-	-	1 (12.5)	-	-	-	-	-	-	-	-	-	1 (12.5)	-
	*S. simulans* (5)	-	-	-	2 (40)	1 (20)	1 (20)	-	-	-	1 (20)	1 (20)	-	1 (20)	-
	*S. lentus* (2)	1 (50)	-	1 (50)	1 (50)	1 (50)	1 (50)	1 (50)	-	-	-	-	-	1 (50)	1 (50)
	*S. warneri* (2)	1 (50)	-	1 (50)	-	-	1 (50)	-	-	1 (50)	-	-	-	0	1 (50)
	*S. epidermidis* (1)	1 (100)	-	1 (100)	-	-	-	-	1 (100)	1 (100)	-	-	-	0	1 (100)
	CoNS Total (29)	3 (10.3)	-	4 (13.8)	3 (10.3)	2 (6.9)	3 (10.3)	1 (3.4)	1 (3.4)	2 (6.9)	1 (3.4)	1 (3.4)	-	10 (34.5)	3 (10.3)
TOTAL		13 (22.4)	-	14 (24.1)	9 (15.5)	18 (31)	15 (25.9)	13 (22.4)	4 (6.9)	2 (3.4)	1 (1.7)	3 (5.2)	-	24 (41.4)	17 (29.3)

^1^ MDR: The NAS isolates that were resistant to three or more subclasses of antimicrobial drugs are defined as MDR isolates. NAS, non-*aureus* staphylococci; CoVS, coagulase-variable staphylococci; CoNS, coagulase-negative staphylococci; AMP, ampicillin; FOX, cefoxitin; PEN, penicillin; CHL, chloramphenicol; FQN, fluoroquinolones (including CIP, ciprofloxacin; ENR, enrofloxacin; LEV, levofloxacin); CLI, clindamycin; ERY, erythromycin; GEN, gentamycin; MUP, mupirocin; RIF, rifampin; SXT, trimethoprim-sulfamethoxazole; SYN, quinupristin-dalfopristin; TET, tetracycline; MDR, multi-drug resistance.

**Table 3 antibiotics-09-00809-t003:** Antimicrobial resistance profiles and distribution of enterotoxin genes in non-aureus staphylococci (NAS) strains isolated from retail chicken meat.

Strains	Isolates ID	Antimicrobial Resistance Profiles	TET-Resistance Genes	MICs (μg/mL)				Staphylococcal Enterotoxin Genes
				TET	OXA	VAN	LZD	
CoVS								
*S. agnetis*	SA1	AMP-PEN -FQN-CLI-ERY-GEN	-	0.5	0.25	1	1	*seh, selj, sem, sep*
	SA2	-	-	0.5	0.25	1	1	*selj, sep, seh*
	SA3	CHL-FQN	-	0.125	0.25	1.5	0.75	*selj, sep, seh*
	SA4	CHL-FQN	-	0.125	0.25	1.5	0.75	*selj*
	SA5	-	-	1	0.125	1	1	*tst1, sep, seh, seo*
	SA6	AMP-PEN-FQN	-	0.125	0.25	1.5	0.75	*sep, seh, sek*
	SA7	AMP-PEN-FQN-GEN	-	0.25	0.5	1	0.5	*sep, seh, seo, sek*
	SA8	AMP-PEN-CLI-ERY	-	0.5	0.25	0.75	0.5	*sep, seh*
	SA9	AMP-PEN-CLI-ERY	-	0.5	0.25	1	0.5	*sep, seh*
	SA10	AMP-PEN-CLI-ERY-GEN	-	0.5	2	1	1	*selj, sep, seh*
	SA11	CHL-FQN	-	1	2	1	0.5	*selj, sep, seh*
	SA12	FQN	-	1	2	0.75	0.75	*sep, seh, seo*
*S. chromogenes*	SC1	FQN-CLI-ERY-TET	*tet*(L)	32	0.25	1	0.38	*tst1, seh, selj, sem, sep*
	SC2	FQN-CLI-ERY-TET	*tet*(L)	32	0.25	1	0.5	*selj, sek*
	SC3	FQN-CLI-ERY-TET	*tet*(L)	32	0.25	1	0.5	*sea, selj, sek*
	SC4	AMP-PEN-FQN-CLI-ERY-SXT-TET	*tet*(L)	16	0.25	1	0.5	*tst1, sek*
	SC5	FQN-CLI-ERY-TET	*tet*(L)	32	0.5	1	1	*tst1, selj, seh*
	SC6	FQN-CLI-ERY-TET	*tet*(L)	32	0.25	0.75	0.5	*tst1, seh, seo*
	SC7	AMP-PEN-CHL-FQN-CLI-ERY-SXT	-	0.125	0.25	0.75	0.5	*tst1, seh, sei, selj*
	SC8	AMP-PEN-CHL-FQN	-	0.125	0.5	0.75	0.5	*tst1, seh, sei, selj, seo*
	SC9	AMP-PEN-CHL-FQN-CLI-ERY	-	0.125	0.25	0.38	0.5	*seh, sei, selj, seo*
*S. hyicus*	SH1	TET	*tet*(L)	16	0.25	1.5	0.75	*sep, seh, seo, sek*
	SH2	TET	*tet*(L)	16	0.25	1.5	0.75	*tst1, seh, seo*
	SH3	TET	*tet*(L)	16	0.25	1	0.75	*tst1, seh*
	SH4	TET	*tet*(L)	16	0.25	1.5	0.5	*tst1*
	SH5	TET	*tet*(L)	16	0.25	1	0.75	*-*
	SH6	TET	*tet*(L)	16	0.25	1	0.75	*sem*
	SH7	TET	*tet*(L)	16	0.25	1	0.5	*sem*
	SH8	TET	*tet*(L)	16	0.25	1	1	*sem*
CoNS								
*S. saprophyticus*	SS1	TET	*tet*(K)	32	1	1	0.5	*-*
	SS2	TET	*tet*(K)	32	1	1.5	1	*-*
	SS3	TET	*tet*(K)	32	1	1	0.5	*-*
	SS4	-	-	0.5	1	1	0.5	*-*
	SS5	-	-	0.25	0.5	1.5	0.75	*-*
	SS6	TET	*tet*(K)	32	0.5	1.5	0.5	*see*
	SS7	TET	*tet*(K)	32	0.5	1	0.5	*see*
	SS8	-	-	0.25	0.5	1	0.5	*-*
	SS9	TET	*tet*(K)	32	1	1	0.5	*-*
	SS10	TET	*tet*(K)	16	0.5	1	0.5	*-*
	SS11	-	-	0.5	0.25	1	0.5	*-*
*S. sciuri*	SI1	-	-	1	1	0.5	0.5	*sell, seo*
	SI2	PEN	-	1	1	0.75	0.5	*sell, seo*
	SI3	-	-	0.25	1	0.5	0.75	*seo*
	SI4	-	-	0.5	0.5	0.5	1	*-*
	SI5	-	-	0.5	0.25	0.75	0.75	-
	SI6	TET	*tet*(K)	16	0.25	0.5	0.5	*-*
	SI7	-	-	0.5	1	0.5	0.5	*-*
	SI8	-	-	0.25	0.5	0.5	0.5	*sell, seo*
*S. simulans*	SM1	RIF-TET	*tet*(M), *tet*(K)	32	0.125	0.5	0.5	*tst1, selj*
	SM2	-	-	0.25	0.25	0.38	0.5	*-*
	SM3	CHL-CLI	-	0.5	0.125	0.38	0.5	*sep, seh*
	SM4	CHL-FQN	-	0.25	0.125	0.5	0.75	*sep*
	SM5	SXT	-	0.5	0.25	0.5	0.5	*-*
*S. lentus*	SL1	TET	*tet*(K)	16	1	1	0.5	*seh, sell*
	SL2	AMP-PEN-CHL-FQN-CLI-ERY	-	1	16	1	0.75	*sell*
*S. warneri*	SW1	-	-	0.25	0.75	1.5	0.5	*tst1, sei, selj*
	SW2	AMP-PEN-CLI-MUP	-	0.5	0.5	0.5	0.5	*seq*
*S. epidermidis*	SE1	AMP-PEN-GEN-MUP	-	0.25	0.125	0.5	0.75	*sep*

VAN, vancomycin; LZD, linezolid.

**Table 4 antibiotics-09-00809-t004:** Mutations in the quinolone resistance determining regions (QRDRs) of *gyrA*, *gyrB*, *parC*, and *parE* in fluoroquinolone-resistant non-*aureus* staphylococci (NAS) isolated from retail chicken meat.

Strains	Isolates ID	MICs (μg/mL)		Mutations in QRDRs			
		CIP	ENR	*gyrA*	*gyrB*	*parC*	*parE*
CoVS							
*S. agnetis*	SA1	16	8	S84L	-	S80L	-
	SA3	8	8	S84L	-	S80L	-
	SA4	16	8	S84L	-	S80L	K465H N488T
	SA6	64	8	S84L	K467R	S80F D84E	D424E K465R N488T N491K
	SA7	32	8	S84L	-	S80L	-
	SA11	4	4	S84L	-	S80L	D424E K465R N488T N491K
	SA12	4	4	S84L	-	S80L	D424E N426K K465R N488T D489E
*S. chromogenes*	SC1	32	128	S84L	-	S80L D84H	-
	SC2	64	128	S84L	-	S80L D84H	-
	SC3	128	128	S84L	-	S80L D84H	-
	SC4	128	128	S84L A132T S162A	-	S80L D84G	-
	SC5	32	128	S84L	-	S80L D84H	-
	SC6	64	128	S84L	-	S80L D84H	-
	SC7	32	128	S84L	-	S80L D84Y	-
	SC8	64	128	S84L	-	S80L D84Y	-
	SC9	32	128	S84L	-	S80L D84Y	-
CoNS							
*S. simulans*	SM4	256	256	S84L A173S	-	D84N	-
*S. lentus*	SL2	32	16	S84L T172A	-	S80L	-

QRDR, quinolone resistance-determining region.

**Table 5 antibiotics-09-00809-t005:** Prevalence of TSST-1, classical SE, and newer SE genes in non-*aureus* staphylococci (NAS) strains isolated from retail chicken meat.

NAS (*n* = Isolates)	No. of SE Genes																			
	*tst1*	*sea*	*seb*	*sec*	*sed*	*see*	*seg*	*seh*	*sei*	*selj*	*sek*	*sell*	*sem*	*sen*	*seo*	*sep*	*seq*	*ser*	*selu*	ND
CoVS																				
*S. agnetis* (12)	1	-	-	-	-	-	-	11	-	6	2	-	1	-	3	11	-	-	-	-
*S. chromogenes* (9)	6	1	-	-	-	-	-	6	3	7	3	-	1	-	3	1	-	-	-	-
*S. hyicus* (8)	3	-	-	-	-	-	-	3	-	-	1	-	3	-	2	1	-	-	-	1
CoVS Total (%)	10 (34.5)	1 (3.4)	-	-	-	-	-	20 (69)	3 (10.3)	13 (44.8)	6 (20.7)	-	5 (17.2)	-	8 (27.6)	13 (44.8)	-	-	-	1 (3.4)
CoNS																				
*S. saprophyticus* (11)	-	-	-	-	-	2	-	-	-	-	-	-	-	-	-	-	-	-	-	9
*S. sciuri* (8)	-	-	-	-	-	-	-	-	-	-	-	3	-	-	4	-	-	-	-	4
*S. simulans* (5)	1	-	-	-	-	-	-	1	-	1	-	-	-	-	-	2	-	-	-	2
*S. lentus* (2)	-	-	-	-	-	-	-	1	-	-	-	2	-	-	-	-	-	-	-	-
*S. warneri* (2)	1	-	-	-	-	-	-	-	1	1	-	-	-	-	-	-	1	-	-	-
*S. epidermidis* (1)	-	-	-	-	-	-	-	-	-	-	-	-	-	-	-	1	-	-	-	-
CoNS Total (%)	2 (6.9)	-	-	-	-	2 (6.9)	-	2 (6.9)	1 (3.4)	2 (6.9)	-	5 (17.2)	-	-	4 (13.8)	3 (10.3)	1 (3.4)	-	-	15 (51.7)
TOTAL	12 (20.7)	1 (1.7)	-	-	-	2 (3.4)	-	22 (37.9)	4 (6.9)	15 (25.9)	6 (10.3)	5 (8.6)	5 (8.6)	-	12 (20.7)	16 (27.6)	1 (1.7)	-	-	16 (27.6)

ND, not detected.

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
