# Peer review of "Species Distribution, Antimicrobial Resistance, and Enterotoxigenicity of Non-*aureus* Staphylococci in Retail Chicken Meat"

_antibiotics, 2020, doi:10.3390/antibiotics9110809_

Round 1
Reviewer 1 Report
1) There are some incorrect values in the first part of Table 1 and in the text above (line 76-77):
S. agnetis (n = 12, 19.4%) → 20,7%
S. chromogenes (n = 9, 14.5%) → 15,5%
S. hyicus (n = 8, 12.9%) → 13,8%
2) At the beginning of chapter 2.3, I recommend using the full name of the antibiotics used, not just their abbreviation.
Author Response
We appreciate the careful review of our manuscript from the Reviewer.
1) There are some incorrect values in the first part of Table 1 and in the text above (line 76-77):
S. agnetis (n = 12, 19.4%) → 20,7%
S. chromogenes (n = 9, 14.5%) → 15,5%
S. hyicus (n = 8, 12.9%) → 13,8%
Response: The values in Table 1 were corrected.
2) At the beginning of chapter 2.3, I recommend using the full name of the antibiotics used, not just their abbreviation.
Response: The abbreviations for antibiotics in section 2.3. were changed to full names as suggested.
Reviewer 2 Report
The study presents the results of an important survey on the antimicrobial resistance and presence in retail chicken meat of staphylococcal superantigens (SAgs) genes including the staphylococcal enterotoxins and other superantigens in Non-aureus Staphylococci. The topic as a high significance for the profile detected and there are not technical comments for improving the quality of the paper. However, the authors should pay attention to the classification of the enterotoxins both in the main text and in the introduction section. The staphylococcal enterotoxins (SEA to SEE; SEG to SEI; SEK; SEM to SET) have been reported as responsible agents for food-borne outbreaks [Regulation (EC) No. 2073/2005; Sato’o et al., 2014, Fetsch et al., 2014], the staphylococcal enterotoxin-like toxins (SElJ; SElL; SElU to SElZ) which are a group of SEs that are not emetic in a primate model, or, their relationship with food poisoning, have yet to be experimentally demonstrated [Macori et al., 2020; Benkerroum et al., 2018; Johler et al., 2015] and the toxic shock syndrome toxin (TSST-1) [Zhao et al., 2016]. For this reasons, the genes sej, sel and seu must be reported as "selj, sell and selu" consistently in the text. Pay attention as well to the italics on the spp. of staphylococci (e.g. Line 71 and L221).
Author Response
Reviewer 2
The study presents the results of an important survey on the antimicrobial resistance and presence in retail chicken meat of staphylococcal superantigens (SAgs) genes including the staphylococcal enterotoxins and other superantigens in Non-aureus Staphylococci. The topic as a high significance for the profile detected and there are not technical comments for improving the quality of the paper.
Response: We appreciate the favorable review of our manuscript and the comments of the Reviewer.
However, the authors should pay attention to the classification of the enterotoxins both in the main text and in the introduction section. The staphylococcal enterotoxins (SEA to SEE; SEG to SEI; SEK; SEM to SET) have been reported as responsible agents for food-borne outbreaks [Regulation (EC) No. 2073/2005; Sato’o et al., 2014, Fetsch et al., 2014], the staphylococcal enterotoxin-like toxins (SElJ; SElL; SElU to SElZ) which are a group of SEs that are not emetic in a primate model, or, their relationship with food poisoning, have yet to be experimentally demonstrated [Macori et al., 2020; Benkerroum et al., 2018; Johler et al., 2015] and the toxic shock syndrome toxin (TSST-1) [Zhao et al., 2016]. For this reasons, the genes sej, sel and seu must be reported as "selj, sell and selu" consistently in the text.
Response: We appreciate the careful review of the reviewer on the classification of enterotoxins. The three enterotoxin-like genes (selj, sell, and selu) in the manuscript were corrected as suggested.
Pay attention as well to the italics on the spp. of staphylococci (e.g. Line 71 and L221).
Response: we have italicized the words.
Reviewer 3 Report
The manuscript presented here reports a study on contamination of retail chicken meat with Non Staphylococcus aureus (NAS) staphylococcal species, resistance phenotypes of the isolates obtained, resistance genes and resistance mutations for the most frequent resistance traits (tetracyclin and flouroquinolones respectively). Furthermore, the isolates were checked for containing genes coding for staphylococccal enterotoxins (SE`s) and toxic shock syndrome toxin TSST-1.
There are not that many reports on NAS on chicken meat until now, in so far the study presented here contributes to fill this gap and is basically worth of being published.
The data on antibiotic resistance traits largely confirm results from previous studies, however, tests for two of critically important antibiotics are lacking (see below). The data on enterotoxin genes and tst are of contribute to a varying picture of this issue.
The following points need to be addressed in a revised manuscript:
- Introduction
- Line 40: reports om mecC and mecB in NAS should be added.
- Line 46: this statement goes far back to the beginning of the 1970ties and better should be omitted.
- Line 51: there are several previous reports on SE`s and TSST-1 in NAS should be referred to.
- Line 58 and following: several of previous reports on SE`s in NAS from meat products stress the potential risk for food poisoning. A statement on previous reports of food poisoning with NAS is desirable, it seems that these cases are very rare so far. The same applies to staphylococcal toxic syndrome with NAS, I aware of one report (Crass and Berdoll, 1986).
- Results
- Line 68: prevalence of NAS (58 among 128 samples) should be mentioned with a more clear wording. Although the methodology used (further processing of 2 – 3 colonies from one selective plate) may not be suitable to answer this question, it would be interesting to know whether different staphylococcal species were detected in one and the same sample.
- Line 88 and following, antibiotic resistance profiles: unfortunately the isolates were not tested for susceptibility against linezolid and vancomycin, two antibiotics which are critically important antibiotics (according to the WHO classification) for treatment of infections in humans. It is highly desirable to retest the 58 isolates (this comparatively low number is really manageable)! According to my view this point is crucial. In case of finding isolates with phenotypic resistance PCR for transferable resistance genes (cfr, optrA and vanA respectively) has to be performed. In particular the risk transfer of genes conferring oxazolidinone resistance to staphylococcal species relevant for infections in humans is of particular interest (e.g. Cuny et al., 2017). Furthermore, it should be explained for which reasons susceptibility against 3 fluoroquinolones (CIP, EN, LEV), for which in staphylococci cross resistance is known, were tested. Nalidixix acid is obsolete for staphylococci.
- Discussion
- There are number of reports (also of more recent ones) on coagulase negative staphylococci as a probable reservoir of transferable resistance genes which should be mentioned.
- Line 172 and the following:: the limit of the study with respect to the lack of demonstration of mecC and mecB should be mentioned.Resistance to erythromycin and to clindamycin is also quite prevalent. Its detection in NAS species which only occasionally occur in humans indicates selection by antibiotics used in livestock such as tylosin. Is this antibiotic still in use in Korea ? Furthermore, a number of isolates exhibits resistance to chloramphenicol, an antibiotic out of use in humans since the early 1970ties. There is, however, cross resistance to florfenicol (mediated by fexA, cfr, optrA). Is florfenicol used in poultry in Korea ?
- Line 284 and following: it should be discussed in more detail that the “newer” SE`s seem to of minor significance of staphylococcal food poisoning, according to my view only on case associated with SHE (she) has been reported so far. Different from data of previously published studies the number of NAS isolates containing genes for “classical” SE`S is low in the sample investigated, this should be discussed also.
Author Response
Reviewer 3
The manuscript presented here reports a study on contamination of retail chicken meat with Non Staphylococcus aureus (NAS) staphylococcal species, resistance phenotypes of the isolates obtained, resistance genes and resistance mutations for the most frequent resistance traits (tetracyclin and flouroquinolones respectively). Furthermore, the isolates were checked for containing genes coding for staphylococccal enterotoxins (SE`s) and toxic shock syndrome toxin TSST-1.
There are not that many reports on NAS on chicken meat until now, in so far the study presented here contributes to fill this gap and is basically worth of being published.
The data on antibiotic resistance traits largely confirm results from previous studies, however, tests for two of critically important antibiotics are lacking (see below). The data on enterotoxin genes and tst are of contribute to a varying picture of this issue.
Response: We appreciate the favorable review of our manuscript and the insightful comments of the Reviewer.
The following points need to be addressed in a revised manuscript:
- Introduction
Line 40: reports on mecC and mecB in NAS should be added.
Response: Sentences on mecB and mecC along with additional references were added in the revised manuscript.
Line 46: this statement goes far back to the beginning of the 1970ties and better should be omitted.
Response: The sentence has been deleted as suggested.
Line 51: there are several previous reports on SE`s and TSST-1 in NAS should be referred to.
Response: Additional references on SEs and TSST-1 in NAS were added in the revised manuscript.
Line 58 and following: several of previous reports on SE`s in NAS from meat products stress the potential risk for food poisoning. A statement on previous reports of food poisoning with NAS is desirable, it seems that these cases are very rare so far. The same applies to staphylococcal toxic syndrome with NAS, I aware of one report (Crass and Berdoll, 1986).
Response: We agree with the reviewer that confirmed cases of SFP caused by CoNS or NAS are extremely rare and early report of CoNS being a cause of TSS/SFP (Crass and Berdoll, 1986) has not been confirmed.
A statement on this point has been added in the revised manuscript.
- Results
Line 68: prevalence of NAS (58 among 128 samples) should be mentioned with a more clear wording. Although the methodology used (further processing of 2 – 3 colonies from one selective plate) may not be suitable to answer this question, it would be interesting to know whether different staphylococcal species were detected in one and the same sample.
Response: We agree with the reviewer that different species of staphylococci could have been isolated from the same chicken meat samples. However, since we picked 2-3 colonies of the most dominant colony type on each selective plate, all the 58 NAS strains used in this study were isolated from different samples.
Line 88 and following, antibiotic resistance profiles: unfortunately the isolates were not tested for susceptibility against linezolid and vancomycin, two antibiotics which are critically important antibiotics (according to the WHO classification) for treatment of infections in humans. It is highly desirable to retest the 58 isolates (this comparatively low number is really manageable)! According to my view this point is crucial. In case of finding isolates with phenotypic resistance PCR for transferable resistance genes (cfr, optrA and vanA respectively) has to be performed. In particular the risk transfer of genes conferring oxazolidinone resistance to staphylococcal species relevant for infections in humans is of particular interest (e.g. Cuny et al., 2017). Furthermore, it should be explained for which reasons susceptibility against 3 fluoroquinolones (CIP, EN, LEV), for which in staphylococci cross resistance is known, were tested. Nalidixix acid is obsolete for staphylococci.
Response: As suggested by the Reviewer, susceptibilities to the two critically important antibiotics (vancomycin and linezolid) were determined by standard Etest method and the MIC values have been added to Table 3 in the revised manuscript. Based on the MIC data, none of the 58 NAS strains exhibited resistance to vancomycin or linezolid. However, several sentences regarding the significance of the vancomycin- and linezolid-resistant staphylococci have been added in Discussion section of the revised manuscript.
The extensive use of fluoroquinolones has resulted in the high prevalence of fluoroquinolone (FQN)-resistant staphylococcus aureus in poultry farms in Korea. Although the cross-resistance has been known, 3 different fluoroquinolones (CIP, EN, and LEV) were used to ensure the FQN-resistance in NAS isolates. Since all the FQN-resistant strains exhibited cross-resistance to the 3 antibiotics, the 3 columns for CIP, EN, and LEV in Table 2 were merged to one FQN column.
Nalidixic acid has been deleted from the manuscript 2 as suggested.
- Discussion
There are number of reports (also of more recent ones) on coagulase negative staphylococci as a probable reservoir of transferable resistance genes which should be mentioned.
Response: Additional statements and references that highlight the role of CoNS in transfer of antimicrobial resistance genes were added in the revised manuscript.
Line 172 and the following:: the limit of the study with respect to the lack of demonstration of mecC and mecB should be mentioned. Resistance to erythromycin and to clindamycin is also quite prevalent. Its detection in NAS species which only occasionally occur in humans indicates selection by antibiotics used in livestock such as tylosin. Is this antibiotic still in use in Korea ? Furthermore, a number of isolates exhibits resistance to chloramphenicol, an antibiotic out of use in humans since the early 1970ties. There is, however, cross resistance to florfenicol (mediated by fexA, cfr, optrA). Is florfenicol used in poultry in Korea ?
Response: The lack of demonstration of mecB and mecC in NAS strains was mentioned in the discussion section of the revised manuscript. Tylosin, chloramphenicol, erythromycin, and clindamycin have not been used in poultry farms in Korea. However, florfenicol is used to treat/prevent respiratory disease in poultry. Occurrence of chloramphenicol resistance phenotype in some NAS isolates might have been affected by the use of florfenicol. This point has been added to the revised manuscript.
Line 284 and following: it should be discussed in more detail that the “newer” SE`s seem to of minor significance of staphylococcal food poisoning, according to my view only on case associated with SHE (she) has been reported so far. Different from data of previously published studies the number of NAS isolates containing genes for “classical” SE`S is low in the sample investigated, this should be discussed also.
Response: This is a good point. Our data indicated that the newer SE genes, especially seh, selj, and sep genes, were more frequent than the classical SE genes among the study strains. Additional statements along with new references have been added in the revised manuscript.
Round 2
Reviewer 3 Report
Nearly all of the points of my review were adressed by the authors. However, I am not quite satisfied with the response to
"There are number of reports (also of more recent ones) on coagulase negative staphylococci as a probable reservoir of transferable resistance genes which should be mentioned".
Demonstration of the same resistance genes in different staphylococcal species of course suggest horizontal transmission. There are studies which go more into depth with respect to the structure of plasmids carrying resistance genes ( e.g. Shen et al., J Antimicrob Chemother 2013, Feßler et al. Plasmid 2018) and to transfewrability (Cuny et al. Vet Microbiol 2017).
Two few further minor points:
The beginning of reference 1 should be corrected: Huebner J
line 159: "reserevoir for..." instead of "route for..."
Author Response
We thank your careful and thoughtful review of the manuscript.
"There are number of reports (also of more recent ones) on coagulase negative staphylococci as a probable reservoir of transferable resistance genes which should be mentioned".
Demonstration of the same resistance genes in different staphylococcal species of course suggest horizontal transmission. There are studies which go more into depth with respect to the structure of plasmids carrying resistance genes ( e.g. Shen et al., J Antimicrob Chemother 2013, Feßler et al. Plasmid 2018) and to transfewrability (Cuny et al. Vet Microbiol 2017).
Response: Additional statements with the suggested references have been added to the revised manuscript.
Two few further minor points:
The beginning of reference 1 should be corrected: Huebner J
line 159: "reserevoir for..." instead of "route for..."
Response: The two points were corrected in the revised manuscript.